# Ratiometric Optical Fiber Dissolved Oxygen Sensor Based on Fluorescence Quenching Principle

**DOI:** 10.3390/s22134811

**Published:** 2022-06-25

**Authors:** Yongkun Zhao, Hongxia Zhang, Qingwen Jin, Dagong Jia, Tiegen Liu

**Affiliations:** 1Key Laboratory of Optoelectronics Information Technical Science, College of Precision Instrument and Opto-Electronics Engineering, Tianjin University, Tianjin 300072, China; ykzhao@tju.edu.cn (Y.Z.); jinqingwen111@ruc.edu.cn (Q.J.); dagongjia@tju.edu.cn (D.J.); tgliu@tju.edu.cn (T.L.); 2School of Information Resources Management, Renmin University of China, Beijing 100872, China

**Keywords:** dissolved oxygen, ruthenium complex, quantum dots, fluorescence quenching, optical fiber dissolved oxygen sensor

## Abstract

In this study, a ratiometric optical fiber dissolved oxygen sensor based on dynamic quenching of fluorescence from a ruthenium complex is reported. Tris(4,7-diphenyl-1,10-phenanthrolin) ruthenium(II) dichloride complex (Ru(dpp)_3_^2+^) is used as an oxygen-sensitive dye, and semiconductor nanomaterial CdSe/ZnS quantum dots (QDs) are used as a reference dye by mixing the two substances and coating it on the plastic optical fiber end to form a composite sensitive film. The linear relationship between the relative fluorescence intensity of the ruthenium complex and the oxygen concentration is described using the Stern–Volmer equation, and the ruthenium complex doping concentration in the sol-gel film is tuned. The sensor is tested in gaseous oxygen and aqueous solution. The experimental results indicate that the measurement of dissolved oxygen has a lower sensitivity in an aqueous environment than in a gaseous environment. This is due to the uneven distribution of oxygen in aqueous solution and the low solubility of oxygen in water, which results in a small contact area between the ruthenium complex and oxygen in solution, leading to a less-severe fluorescence quenching effect than that in gaseous oxygen. In detecting dissolved oxygen, the sensor has a good linear Stern–Volmer calibration plot from 0 to 18.25 mg/L, the linearity can reach 99.62%, and the sensitivity can reach 0.0310/[*O*_2_] unit. The salinity stability, repeatability, and temperature characteristics of the sensor are characterized. The dissolved oxygen sensor investigated in this research could be used in various marine monitoring and environmental protection applications.

## 1. Introduction

Determination of the dissolved oxygen (DO) concentration is essential in aquaculture, marine monitoring, and industrial and agricultural production [1,2,3,4]. As a result, considerable efforts have been made over the years to develop a DO sensor that is accurate, efficient, stable, and practical, and has a rapid response. The current DO measurement methods are mainly classified into chemical, electrochemical and optical [5,6,7,8,9]. In chemical DO sensing, Winkler titration is the most accurate test method, but the procedures are complicated, making it difficult to realize in real-time monitoring [10]. Amperometric devices based on the oxidation-reduction reaction between the Clark electrode and oxygen molecules in water are commonly used in electrochemical method-based sensors [11,12]. However, the electrode will keep aging, and the sensor needs frequent calibration. Meanwhile, the sensor is susceptible to electromagnetic interference and consumes the oxygen of the analyte in the electrochemical reaction. Optical fiber sensors have been widely used in medical diagnosis, chemical analysis, marine and environmental monitoring and other fields. It has great potential in gas monitoring [13,14,15]. Various gas sensors based on different methods have been prepared.

Collisional quenching [16,17,18,19,20] between oxygen molecules and fluorescence or phosphorescence molecules in a support matrix is the basis for most optical fiber DO sensors. It has been demonstrated that optical fiber chemical sensors have superiorities over other DO sensors, such as ease of operation, cost-effectiveness, and anti-electromagnetic interference. Several dyes have been used for luminescence-based oxygen sensors, such as polycyclic aromatic hydrocarbons, fullerenes, metal-ligand complexes of ruthenium(II), iridium(III), osmium, rhenium, platinum [21,22], and metalloporphyrin. Ruthenium(II) complexes [23,24,25,26,27,28] have a long fluorescence time, strong visible light absorption, and high metal-ligand charge transfer efficiency; thus, they are widely used in DO sensor detection.

Conventional luminescence intensity measuring methods suffer from many weaknesses, such as inhomogeneous distribution of dyes, reflection background and drifts of the excitation light sources. The luminescence lifetime-based measurement is superior to intensity measurement as it is unaffected by the light source. However, the lifetime-based measurement has a relatively low number of available sensing materials and changes the luminescence lifetime after interaction with the analyte [29]. The ratiometric method [30,31,32,33] can be a choice based on the two different luminophores that exhibit different performance characteristics on varying *O*_2_ concentrations. We employed the ratiometric approach to identify oxygen sensors as it generally yields more robust sensing systems that are less influenced by difficulties associated with non-analyte-induced intensity variations.

The sensor’s analytical performance was studied in detail. The results indicate that this study’s ratiometric DO sensor has wide application prospects in marine monitoring and environmental protection.

## 2. Theory Section

Oxygen has a quenching effect on fluorescence produced by some fluorescent substances, resulting in a decrease in fluorescence intensity and a reduction in fluorescence life. The purpose of an optical fiber DO sensor is to detect oxygen concentration by using a fluorescent substance as an indicator to generate fluorescence under light excitation of a specific wavelength, and oxygen has a quenching effect on fluorescence through the detection of the indicator’s degree of fluorescence. As presented in Figure 1a, we use ruthenium complex and QDs to coat the end of the optical fiber. When the excitation light enters the optical fiber, the fluorescent substance absorbs light of a specific wavelength. Its electrons gain energy, become excited, and release energy to return to the ground state by emitting fluorescence of a higher wavelength. The emission light is reflected out from the optical fiber. The fluorescence quenching reaction occurs at the interface of the membrane when the DO sensor is put into water and emitted by the light source.

The Stern–Volmer equation [34] can describe the relationship between fluorescence intensity or fluorescence lifetime and oxygen concentration:(1)$$\frac{I_{0}}{I}=1+K_{SV}[O_{2}]$$
where $I_{0}$ and I denote the fluorescence intensity of the luminophore in the absence and presence of oxygen; $K_{SV}$, the Stern-Volmer quenching constant(for specific fluorescence quenchers, $K_{SV}$ is fixed); and $\left[O_{2}\right]$, the oxygen concentration. For an ideal case of a homogeneous environment, a plot of the $I_{0}/I$ versus [*O*_2_] will be a straight line with an intercept at 1 and a slope of $K_{SV}$.

Although the sensor can detect DO using only the ruthenium complex, as the light source significantly fluctuates, the fluorescence intensity of the ruthenium complex will significantly change, making the fluorescence intensity inconsistent with the previous calibration, resulting in inaccurate DO measurements. Thus, optical fiber DO sensor using only ruthenium complex is more or less affected by light source fluctuations, indicator attenuation, and other external environmental factors, so we employed the ratiometric method for the oxygen sensor detection. Figure 1b presents the QDs fluorescence emission when the light source excites it. An LD light source with a center wavelength of about 405 nm was used to excite CdSe/ZnS QDs. They emitted fluorescence around 530 nm, and the oxygen concentration did not affect its excitation. Thus, we chose QDs as reference materials. Figure 1c presents the quenching effect of the ruthenium complex. It can be seen that the fluorescence intensity changed accordingly with the change in oxygen concentration. The Stern–Volmer equation is briefly popularized, which is expressed as follows:(2)$$\frac{P_{0}}{P}=\frac{I_{0}/I_{QDs0}}{I/I_{QDs}}=1+K_{SV}[O_{2}]$$
where $I_{QDs0}$ and $I_{QDs}$ denote the fluorescence intensity of QDs in the absence and presence of oxygen.

## 3. Experimental Section

### 3.1. Reagents and Materials

Chemical reagents required for this experiment include ethyl cellulose (EC; Tianjin Botai Yida Biotechnology Co., Ltd., Tianjin, China), toluene (Tianjin Yuanli Chemical Co., Ltd., Tianjin, China), tris(4,7-diphenyl-1,10-phenanthrolin) ruthenium(II) dichloride complex (purity 98%, purchased from Shanghai Mairuier Chemical Technology Co., Ltd., Shanghai, China), oilsoluble CdSe/ZnS QDs solution (5 mg/mL, purchased from Suzhou Xingshuo Nano Technology Co., Ltd., Suzhou, China), absolute ethanol (purity ≥ 99.8%), nitrogen and oxygen (purity 99.999%, Tianjin Boliming Technology Co., Ltd., Tianjin, China), and a large amount of deionized water. All chemical reagents were used without further purification.

### 3.2. Preparation of Sensitive Membrane

As presented in Figure 2a, the support matrix material EC used in the sensor was prepared by mixing EC (3 g), ethanol (6 mL), and toluene (20 mL) to produce the encapsulation matrix (solution A). As presented in Figure 2b, the sensitive dye (solution B) was prepared by dissolving 2 mg of ruthenium complex into 10 mL of ethanol. The reference dye (solution C) was prepared by dissolving 2 mg of CdSe/ZnS QDs into 10 mL of toluene, as presented in Figure 2c. The sensing solutions were then prepared by mixing solution A (1 mL) and solution B (1 mL) into solution C (3 mL), as presented in Figure 2d. The thickness of the membrane is not related to the concentration of the ruthenium complex. When the optical fiber was coated, the composite solution was absorbed by a syringe and added to the end of the optical fiber with one drop in every 2–3 min. This procedure was repeated 2–3 times. Subsequently, the optical fiber was dip-coated about 30 times using a dipping-pulling coating machine. Finally, the coated optical fiber was placed in an ambient environment for 1 week. Changes in the film thickness were expected, but the difference was not large; furthermore, the effect on the experimental results was very small. The coating procedure was also the same, and the coating thickness was between 400–500 μm.

### 3.3. Instrumentation

Figure 3 presents the schematic and physical picture of the experimental system. The excitation light was provided by an LD light source (MW-GX-405/150 mW, Changchun Leishi Optoelectronics Technology Co., Ltd., Changchun, China). The composite material was deposited on a 1000-μm-diameter infrared plastic optical fiber (FC/PC-BGUV1000-0.6, Beijing Technology Co., Ltd., Beijing, China). The excitation light entered the oxygen-sensitive membrane through one end of the Y-type fiber (UV400, Beijing Shouliang Technology Co., Ltd., Beijing, China). A custom vacuum pump (Suzhou Partner Experimental Equipment Co., Ltd., Suzhou, China) was used to change the oxygen concentration in gas; 500-mL Porous cylinder (Tianjin Tengyin Trading Co., Ltd., Tianjin, China) was used to change the DO concentration in water. After fluorescence excitation, the light was emitted from the other end of the Y-type fiber, and the relative luminescence intensity was measured using an Ocean Optical USB4000 miniature fiber spectrometer (Shanghai Bose Intelligent Technology Co., Ltd., Shanghai, China). The AS720 DO meter (Shanghai Azuwang Trading Co., Ltd., Shanghai, China) was used to measure the oxygen concentration.

## 4. Results and Discussions

### 4.1. Optimal Ruthenium Complex Concentration

We conducted comparative studies on optical fiber DO sensors prepared with four different ruthenium complex concentrations. The effect of the different concentrations of ruthenium complex on sensor sensitivity was verified by obtaining the ruthenium complex fluorescence intensity in saturated nitrogen and oxygen solutions and quenching ratio. Figure 4 demonstrates that the ruthenium complex with different concentrations fluctuates under the same excitation conditions. Its fluorescence quenching ratio is the largest when the concentration is 0.2 mg/mL. The quenching ratio slightly decreases as the ruthenium complex concentration increases; this is due to molecular aggregation, resulting in fluorescence quenching. The ruthenium complex with a concentration of 0.2 mg/mL was selected for the following experiments to improve the sensor’s sensitivity.

### 4.2. Response Curve

The oxygen-sensitive dye and the reference dye were coated, respectively, and the two sensing fibers were tested in gaseous oxygen to measure their fluorescence intensity. Figure 5 demonstrates that the fluorescence intensity of the ruthenium complex is the highest in the anoxic environment as the oxygen concentration increases, the fluorescence intensity gradually decreases, and the quenching effect becomes obvious. The QDs fluorescence intensity did not significantly change under different oxygen concentrations. It remained stable, indicating its suitability for use as a reference dye for ratiometric optical fiber oxygen sensor. Figure 6a presents the fluorescence response spectrum of the sensor made of composite sensitive materials in gaseous oxygen. QDs and ruthenium complexes at about 530 and 615 nm produce two emission peaks; as the oxygen concentration increases, fluorescence intensity gradually decreases due to the ruthenium complex fluorescence quenching, and the QDs fluorescence intensity is in numerical stability. Subsequently, we measured the fluorescence response spectra of the sensor in an aqueous solution, as presented in Figure 6b. As the oxygen concentration increases, the fluorescence intensity of the ruthenium complex gradually weakens, whereas the fluorescence intensity of QDs fluctuates within a certain range.

### 4.3. Sensitivity and Linearity

We measured the peak fluorescence intensity of QDs at 530 nm and the peak fluorescence intensity of ruthenium complex at 630 nm, respectively. In addition, we drew the Stern–Volmer equation curve using Equation (1). Figure 7 presents the changes in the peak value under the separate action of QDs and ruthenium complex. It can be seen that the ruthenium complex coating alone has a good correlation with the linearity up to 96.29%, and the sensitivity calculated reaches 0.0654/[*O*_2_] unit.

Figure 8 presents the fluorescence quenching of the ratiometric optical fiber sensor prepared using a composite-sensitive solution. It can be seen from the scatterplot that the ruthenium complex fluorescence intensity has a certain correlation with the oxygen concentration, whereas the QDs fluorescence intensity is basically stable. Figure 8b presents the Stern–Volmer diagram of the ruthenium complex drawn by Equation (1) in gaseous oxygen, the linearity can reach 98.56% and the sensitivity can reach 0.0660/[*O*_2_] unit. Figure 8d presents the Stern–Volmer diagram of the ruthenium complex in an aqueous solution. The linearity can reach 99.57%, and the sensitivity can reach 0.0320/[*O*_2_] unit. The experimental results indicate that the sensitivity of measuring DO is lower in an aqueous environment than that in a gaseous environment. This is due to the uneven distribution of oxygen in aqueous solution and the low solubility of oxygen in water, so the contact area between the ruthenium complex and oxygen in solution is too small, resulting in a less severe fluorescence quenching effect than that in gaseous oxygen.

Based on Equation (2), Figure 8 was further optimized to obtain the Stern–Volmer equation curve of ratiometric optical fiber sensor under different oxygen concentrations. As presented in Figure 9, the linearity in the gas environment can reach 99.62%, the sensitivity can reach 0.0683/[*O*_2_] unit, and the linearity in an aqueous environment can reach 99.62%; moreover, the sensitivity can reach 0.0310/[*O*_2_] unit. Compared to the use of only the ruthenium complex, as presented in Figure 5a, the ratiometric oxygen sensor has better linearity and about the same sensitivity. When the light source fluctuates or the indicator greatly decays, the calibration of the sensor using only ruthenium complex will be inconsistent with the previous, resulting in an inaccurate measurement. However, ratiometric method-based oxygen sensors do not have such concerns. Table 1 presents a comparison of different ratiometric DO sensors. The studied sensor is easy to fabricate and has a good linear Stern–Volmer calibration plot and moderate sensitivity. The salinity stability, repeatability, and temperature characteristics of the sensor are good.

### 4.4. Effect of Temperature

We introduce the variation coefficient as a reference index for the fluctuation of experimental results,(3)$$C\cdot V=\frac{\sigma }{\mu }\times 100\%$$
where σ is the data standard deviation, and *μ* is the data mean. As can be seen from Figure 10a, the DO meter readings in the saturated nitrogen below 0.30, can approximate nitrogen saturated solution. Figure 10b presents the fluorescence intensity of the ruthenium complex and the QDs fluctuated within a constant range; the variation coefficients of the two substances were 0.68% and 3.92%, which indicated that the fluorescent dyes in saturated nitrogen had better stability. Figure 10c,d demonstrate that the fluorescence intensity of the DO meter and ruthenium complex has a large fluctuation in oxygen saturated solution; as the temperature increased, the solubility of oxygen in water decreased, so the quenching effect decreased. Therefore, the ruthenium complex fluorescence intensity increased as the temperature increased, and the variation coefficient of QDs fluorescence intensity was 5.54%.

### 4.5. Effect of Salinity

We prepared seven different concentrations of artificial seawater and calculated the quenching ratio to evaluate the effect of salinity on the proposed sensor. As presented in Figure 11, the fluorescence quenching ratio is stable at about 3.0, so the salinity stability is good.

### 4.6. Repeatability

The repeatability of DO sensors is an important analytical figure of merit. An experiment was conducted to evaluate the proposed sensing fiber by putting the same sensing fiber in fully deoxygenated (pure N_2_ saturated) and fully oxygenated (pure *O*_2_ saturated) water for 13 days. Nonlinear error is introduced to measure the fluctuation,
(4)$$\gamma_{L}=\pm \frac{\Delta L_{\max}}{Y_{FS}}\times 100\%$$
where $\Delta L_{\max}$ denotes the maximum difference between the collected data and the mean, $Y_{FS}$ is the full-scale value of the collected data. The test results are presented in Figure 12. The maximum nonlinear error of the sensor is 3.81% over some time, with good repeatability.

## 5. Conclusions

The integration of the ratiometric method with an oxygen sensor was achieved in this study. The integrated oxygen sensor was capable of precisely detecting oxygen concentration in a gaseous environment and aqueous solution. The composite film deposited on the plastic fiber end exhibited good temperature stability, salinity stability, and repeatability. The ratiometric optical fiber sensor exhibited good linearity (99.62%), moderate sensitivity, 0.0310/[*O*_2_] unit. The ratiometric-based oxygen sensor outperforms the conventional intensity-based oxygen sensor, with the advantages of insensitivity to light source fluctuations, indicator attenuation and other external environmental factors. This work is useful for oxygen sensor fabrication and DO detection in water quality monitoring.

## Figures and Tables

**Figure 1 sensors-22-04811-f001:**
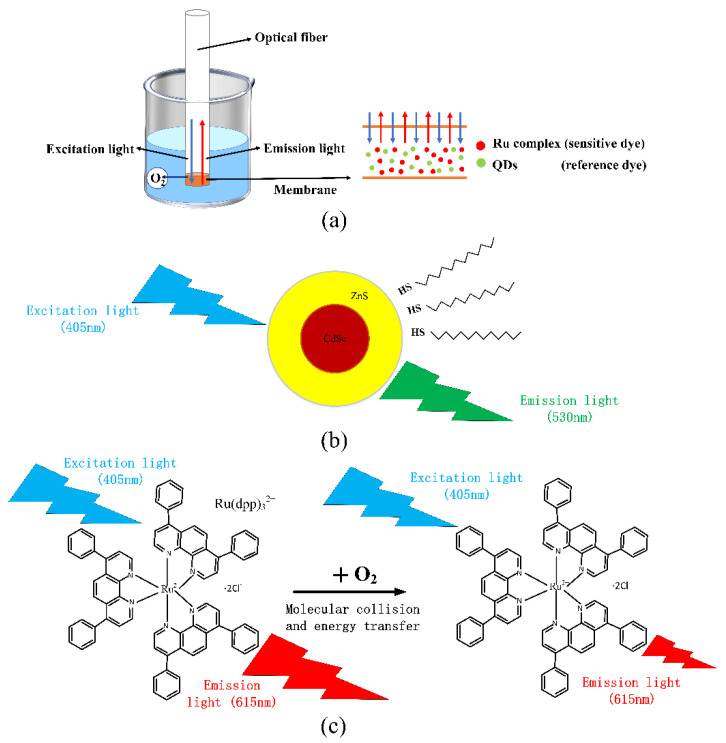
Schematic diagram of the principle of the ratiometric DO sensor (**a**) fluorescence quenching of optical dissolved oxygen sensor (**b**) excitation of QDs (**c**) principle of fluorescence quenching effect.

**Figure 2 sensors-22-04811-f002:**
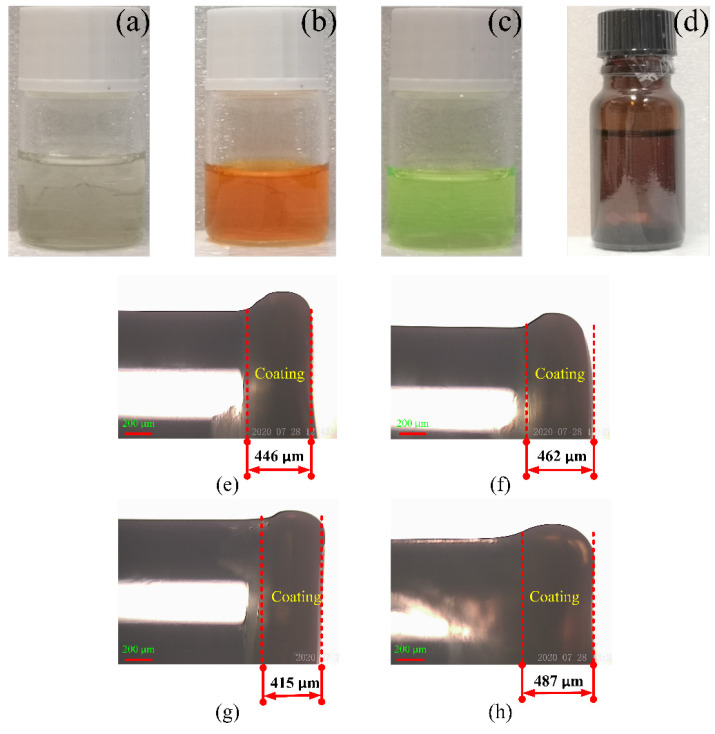
Chemical reagent and coating images of four optical fibers with different ruthenium concentrations (**a**) carrier matrix, (**b**) oxygen sensitive dye, (**c**) reference dye, (**d**) DO composite sensitive solution, (**e**) 0.1, (**f**) 0.2, (**g**) 0.3, and (**h**) 0.4 mg/mL.

**Figure 3 sensors-22-04811-f003:**
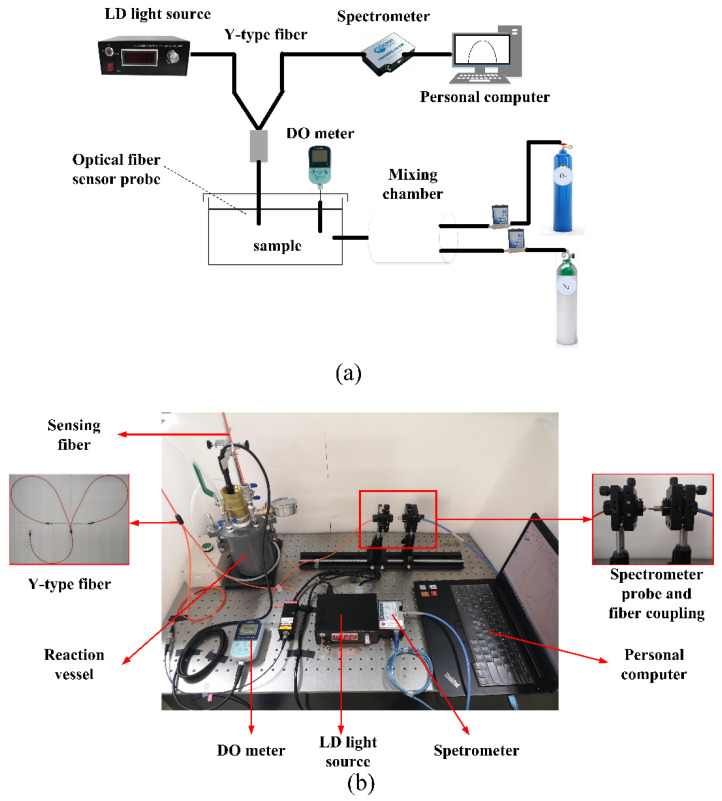
(**a**) Schematic diagram of the experimental setup (**b**) physical picture.

**Figure 4 sensors-22-04811-f004:**
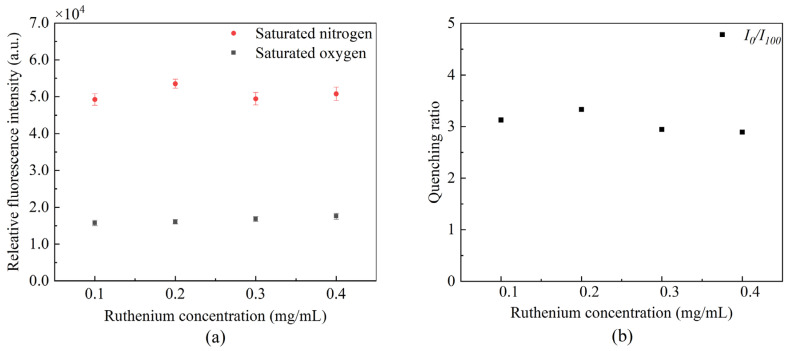
Fluorescence of ruthenium with different concentrations (**a**) fluctuation of fluorescence peak intensity, and (**b**) influence of quenching ratio.

**Figure 5 sensors-22-04811-f005:**
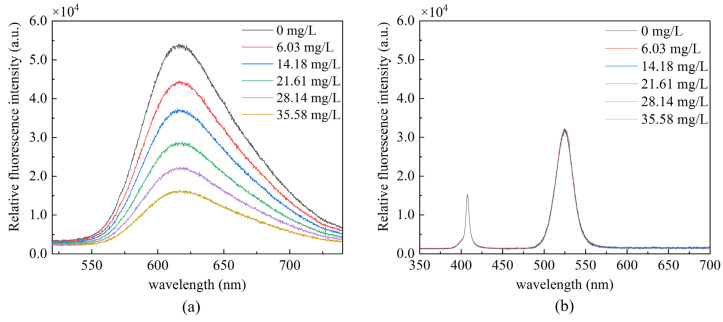
Fluorescence response in gaseous oxygen (**a**) ruthenium complex (**b**) QDs.

**Figure 6 sensors-22-04811-f006:**
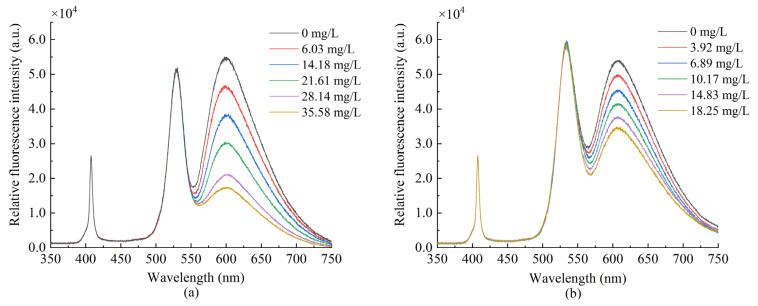
Fluorescence emission spectra of sensor prepared with composite dyes in (**a**) gaseous oxygen (**b**) aqueous solution.

**Figure 7 sensors-22-04811-f007:**
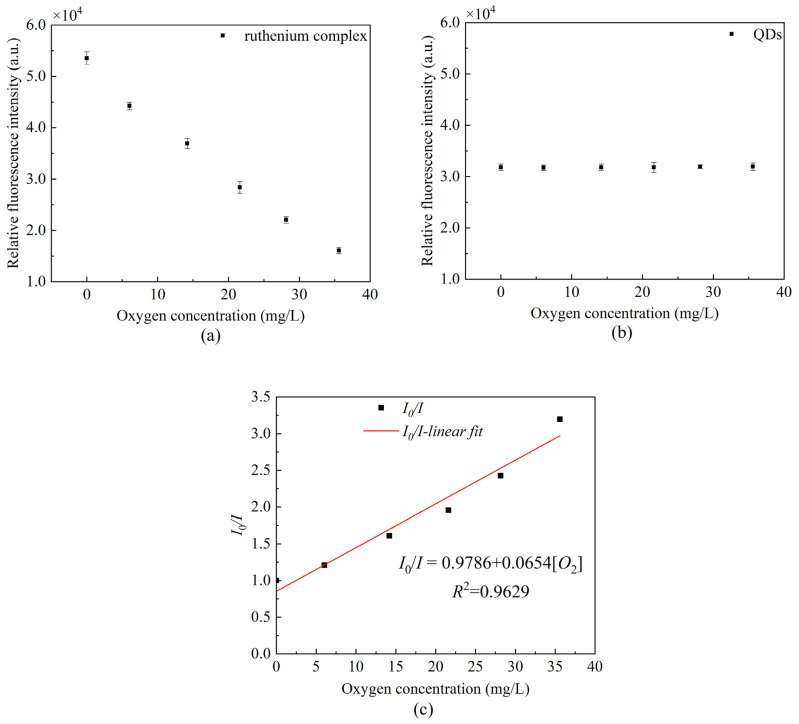
Fluorescence intensity at different oxygen concentrations (**a**) peak fluorescence intensity of ruthenium complex, (**b**) peak fluorescence intensity of QDs, and (**c**) Stern–Volmer equation curve in the presence of ruthenium complex alone.

**Figure 8 sensors-22-04811-f008:**
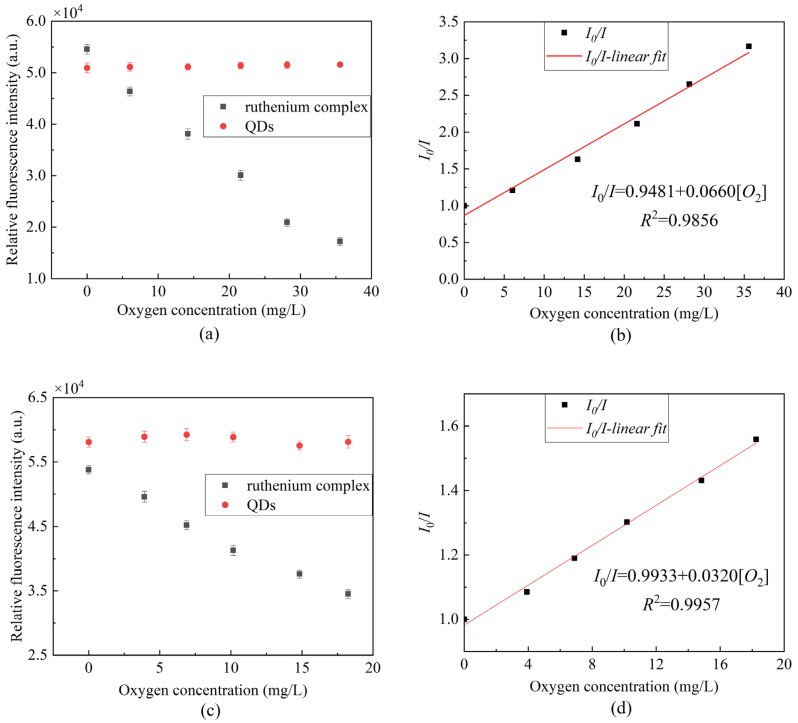
Fluorescence response at different oxygen concentrations (**a**) peak fluorescence intensity of ruthenium complex and QDs in gaseous oxygen, (**b**) Stern–Volmer equation curve of gaseous oxygen in the presence of mixed ruthenium complex and QDs, (**c**) peak fluorescence intensity of ruthenium complex and QDs in aqueous solution, (**d**) Stern–Volmer equation curve of aqueous solution in the presence of mixed ruthenium complex and QDs.

**Figure 9 sensors-22-04811-f009:**
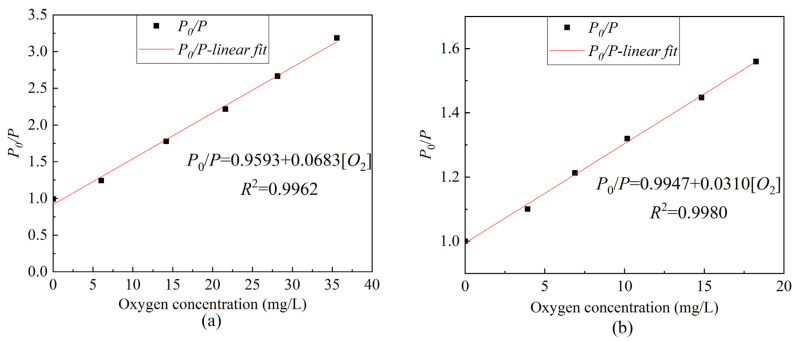
The ratiometric Stern–Volmer equation curve of composite sensitive film under different oxygen concentrations (**a**) gaseous oxygen environment, (**b**) dissolved oxygen environment.

**Figure 10 sensors-22-04811-f010:**
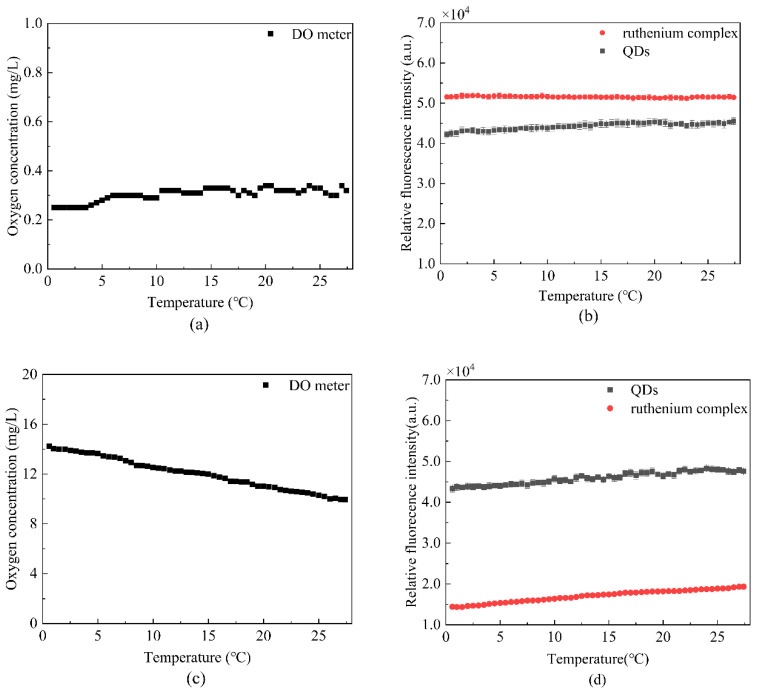
Temperature stability test (**a**) the oxygen concentration of saturated nitrogen solution measured by DO meter at different temperatures, (**b**) the fluorescence intensity fluctuations of ruthenium complex and QDs in saturated nitrogen solution at different temperatures, (**c**) the oxygen concentration of saturated oxygen solution measured by DO meter at different temperatures, and (**d**) the fluorescence intensity fluctuations of ruthenium complex and QDs in saturated oxygen solution at different temperatures.

**Figure 11 sensors-22-04811-f011:**
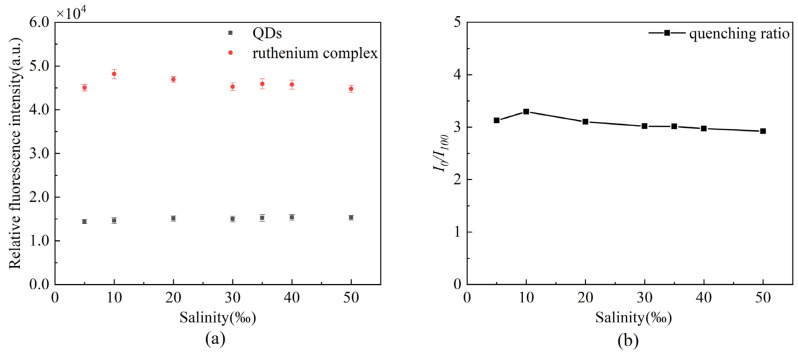
Stability of the oxygen sensor under different salinities (**a**) changes in fluorescence intensity of the two fluorescent dyes, (**b**) changes in fluorescence quenching ratio.

**Figure 12 sensors-22-04811-f012:**
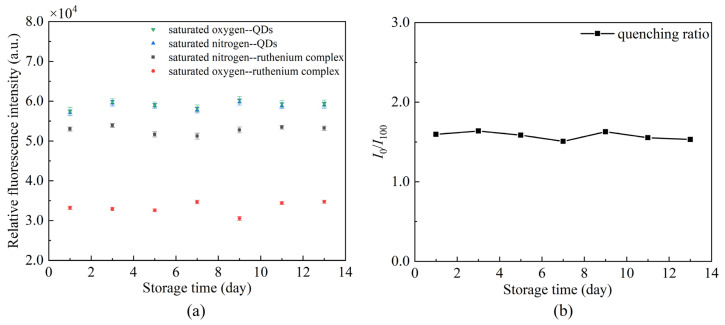
Effects of storage time on the same sensing fiber (**a**) changes of fluorescence intensity, (**b**) changes of quenching ratio.

**Table 1 sensors-22-04811-t001:** Comparison of different ratiometric dissolved oxygen sensors. λ_exc_ and λ_em_ are spectral peaks of excitation and emission, respectively; SVP means Stern-Volmer plot; I_R_ = I_0_/I_100_ is the ratio of fluorescence intensity at 0% *O*_2_ and 100% *O*_2_.

*O*_2_-Sensitive Dye	*O*_2_-Reference Dye	λ_exc_	λ_em_	I_R_	Comments	Ref.
**PtOEPK**	OEP	514 nm	750 nm (sensitive) 620 nm (reference)	-	Good linearity; minimal photobleaching and leaching.	[24]
**Ru(phen)_3_**	NBD-PE	450 nm	600 nm (sensitive) 510 nm (reference)	~3	Phospholipid vesicle sensors; nanometer sized.	[25]
**Ru(dpp)_3_Cl_2_**	Coumarin6	450 nm	608 nm (sensitive) 498 nm (reference)	~1.5	Fibrous structure by PMMA; good mechanical strength.	[26]
**PtTEPP**	CdSe QDs	405 nm	650 nm (sensitive) 515 nm (reference)	~6.5	Linear SVP; intensity method.	[28]
**Ru(dpp)_3_^2+^**	CdSe/ZnS QDs	405 nm	615 nm (sensitive) 530 nm (reference)	~3.33	Good linear SVP; ratiometric approach shows good antijamming capability.	This study

## Data Availability

Not applicable.

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
