# Peer review of "Ratiometric Optical Fiber Dissolved Oxygen Sensor Based on Fluorescence Quenching Principle"

_sensors, 2022, doi:10.3390/s22134811_

Round 1
Reviewer 1 Report
In this submitted manuscript, Dr. Zhang and co-authors studied an optical fiber that dissolves Ru(dpp)3-based sensor to detect oxygen in both gas phase and aqueous solution. This ratiometric detection technique shows a good linear Stern-Volmer plot in the 0-18.25 mg/L range with a sensitivity of 0.031/[O2] unit, indicating good applications in marine monitoring and environmental protection.
While it is important to develop efficient and accurate detecting methods for oxygen in both the gaseous phase and aqueous solution, the manuscript in its current form has weaknesses in clear and substantial illustration of the importance of their work, and the soundness of their conclusions drawn from experimental observations. The language should also be thoroughly revised. Based on that, I would recommend reconsideration after major revisions.
Criticisms include:
1. One of the selling points of this report should be the ratiometric detection approach for oxygen in both the gaseous phase and aqueous solution. However, the “ratiometric” was not sufficiently highlighted and explained in the manuscript. The authors should provide more detailed discussions on how this optical fiber sensor can ratiometrically detect oxygen, and the extraordinary advantage of this technique compared with others.
2. No change can be seen from Fig 1c before and after molecular collision and energy transfer. The authors should revise it to clearly convey the information that is intended to communicate.
3. In Fig 10a, the Y-axis can use the range of 0-1 mg/L instead of 0-10 mg/L to clearly show the change of oxygen concentration along with the temperature. Using 0-10 mg/L is too wide and makes the changes sightless.
4. The conclusion section is stuffless and the authors should strengthen it with a systematic summary of the work reported here and highlight the importance of potential applications.
5. The authors’ information, like their affiliations, was missed.
6. The language should be revised and improved completely. In the current version, there are many fragments gathered together to form a sentence, which is grammatically incorrect. The comma could not connect two fragments to constitute a complete sentence, and it is highly recommended for the authors to seek help from a native speaker to improve the language. Some of the examples are listed below.
The word “and” was missed before “the doping concentration” on line 12. As in English, the fragments without suitable “connecting words” cannot form a complete and correct sentence. The same comments for sentences on lines 19, 32, 43, and many others.
The manuscript should be written in the third person rather than the first. For example, on lines 125-128, it would be more professional to write as “When the optical fiber was coated, the composite solution was absorbed by a syringe and added to the end of optical fiber with one drop in every 2-3 minutes. It was repeated 2-3 times and then dip-coated by a dipping-pulling coating machine for about 30 times”.
Author Response
The response letter and revised manuscript are uploaded. Please see the attachment.

Reviewer 2 Report
The article reported by Yongkun et al. entitled “ratiometric optical fiber dissolved oxygen sensor based on fluorescence quenching principle” present the experimental validation of oxygen sensor using fluorescence material integrated optical fiber. The result exhibits a good linear relation between the fluorescence intensity concentration of oxygen and exhibits good sensitivity of 0.0310/O2. All in all, this is a good piece of work supported experimentally validated, and optimized in terms of doping concentration. The article is a good match to the goal of Sensors, however, there are some concerns that need to be addressed:
1. The state of the art can be improved following the latest review article on gas sensors and discussing the potential of optical fiber in gas monitoring in a few lines. The author may find the following article useful: 10.1016/j.sna.2022.113455, 10.1002/aisy.202100067, 10.3390/s21030731
2. Authors are advised to cite the original article used for equations.
3. How does the author deposits material on tip of the fiber, my concern is regarding its adhesion. Does it require any pre-functionalization of the fiber surface?
4. Why is it required to keep the sensing head in the ambient environment for 1 week? So basically, the experiment was initiated after 1 week of deposition? I’ll suggest the author consider studying this feature of the sensor in the future to study the response of the sensor right after deposition.
5. Fig. 3, can the author show the actual experimental chamber? Also, a detailed discussion is required over the experimental setup in terms of the range of wavelength, type of source, type of source, chamber size, etc. technical detail is a must to reproduce the experiment.
6. Authors are advised to use high-quality pictures.
7. Conclusion needs to be rewritten, as important information is missing.
8. Presentation and technical merit of the work seem quite satisfactory for the reader but grammar and syntax need to be improved.
Author Response

(The authors gave the same response as above.)

Round 2
Reviewer 2 Report
The author has addressed all the raised concerns positively, hence I recommend its publication in Sensors. Good luck